# The Spatial Effect of Financial Innovation on Intellectualized Transformational Upgrading of Manufacturing Industry: An Empirical Evidence from China

**Juanmei Zhou [1,\*], Fenfang Cui [1] and Wenli Wang [2]**

1   College of Economics and Management, North University of China, Taiyuan 030051, China; cuifenfang001@163.com
2   School of Economics and Management, Taiyuan University of Science and Technology, Taiyuan 030051, China; wlwang@tyust.edu.cn
\*   Correspondence: juanmei.zhou@yahoo.com

**Abstract:** Financial innovation is a powerful source to promote the intellectualized transformation and upgrading of the manufacturing industry. At present, the influence mechanism behind the impact of financial innovation on the manufacturing industry is still unclear and lacks quantitative analysis. This paper selects the representative indicators to study the spatial effect of financial innovation on the intelligent transformation and upgrading of the manufacturing industry based on China's provincial panel data from 2008 to 2019 by using entropy method, the DEA model, and the spatial econometric model. The results show the following: (1) the overall level of the intellectualized transformation and upgrading of the manufacturing industry has gradually improved, but the spatial distribution is uneven, which has a strong spatial correlation with financial innovation; (2) according to the estimation of the spatial Dobbin model, financial innovation has a significant positive spillover effect on the intellectualization of the manufacturing industry, not only in the region in which it is located, but also in adjacent regions, and the former is greater than the latter; (3) consumption demand, capital allocation, and technological innovation are the three main mediating mechanisms. Finally, according to the findings, we put forward some suggestions and conclusions. This paper provides a positive reference for further promoting the intelligent upgrading of the manufacturing industry from the perspective of financial innovation.

**Keywords:** financial innovation; manufacturing; intellectualization; capital allocation

## 1. Introduction

The manufacturing industry is the main body of a national economy and the foundation of a country. In the recent digital era, the transformation of China's manufacturing industry has achieved remarkable results, and the upgrading of the industrial structure has accelerated. According to statistics, in 2012, the added value of China's manufacturing industry was 16.98 trillion yuan, up 56.65% year-on-year by 2020. However, compared with European and American manufacturing powers, there is still a significant gap in China's manufacturing industry, and the level of intelligence and technical competitiveness need to be further improved [1]. At the same time, faced with serious problems such as enhanced resource and environmental constraints, and rising labor costs, traditional manufacturing enterprises urgently need to transform to intellectualization [2]. In this context, the realization of intelligent transformation and the upgrading of the manufacturing industry is the direction of future development, and the development of the real economy cannot be separated from the support of finance. Among them, financial innovation is the key link to promote the transformation of the manufacturing industry [3–5]. As the "blood" of the real economy, on the one hand, financial innovation can grasp the direction, scale, and quality of investment and financing, guide resources to the intelligent manufacturing

field with a perfect industrial foundation and support the intelligent transformation and upgrading of the manufacturing industry [3]. On the other hand, financial innovation, with its unique low-cost, high-speed liquidity and spillover effect, guides other economic resources to lean towards regions with a high economic development level, low financing cost, and advanced manufacturing level, so as to optimize the allocation of resources [4,5]. At present, China's manufacturing industry is in the critical period of transformation and upgrading, and scholars and government departments are paying more and more attention to the role of financial innovation in the real economy. Will financial innovation promote the intelligent transformation of the manufacturing industry? If there is a positive impact, how to help improve the core competitiveness of the manufacturing industry? Answering these questions is of great value for exploring a new path for financial innovation and supporting the structural optimization and high-quality development of the manufacturing industry.

Judging from the existing literature, the research on financial innovation and manufacturing transformation mainly focuses on the following aspects. Firstly, whether financial innovation affects the intellectualized transformation and upgrading of manufacturing industry. Scholars hold diverse opinions. Through the information optimization of the financial market, financial innovation will reduce financing costs and liquidity risks, expand financial services, and promote capital accumulation, thus further promoting the intelligent upgrading of the manufacturing industry [6–8]. Meanwhile, some scholars hold that the risks caused by financial innovation will inhibit the transformation of the manufacturing industry [9,10]. Dy et al. [9] believe that the correlation mechanism of external financing of financial institutions is not significant, which will not support the upgrading of the manufacturing industry. Neuenkirch and Nockel [11] point out that financial innovation will lead to the unstable development of the financial industry and aggravate the regional liquidity risk, which is unfavorable for the transformation of the manufacturing industry. Secondly, what is the specific action path towards the intellectualization of the manufacturing industry? There are many studies on this aspect in the existing literature. Manufacturing enterprises use technological innovation [12,13], human capital accumulation [14,15], government incentives [16–18], and other means to improve the success rate of product R&D so as to promote the intelligent manufacturing of the traditional manufacturing industry and to form a stronger supply chain than before. During the course, finance promotes knowledge spillover through industrial agglomeration, affects cost surplus and income surplus, pushes to achieve the state of collaborative innovation coupling between the financial industry and the manufacturing industry, and gradually jumps to the high end of the value chain. Simultaneously, some scholars explore the path of financial innovation to promote the upgrading of the manufacturing industry from an empirical perspective. Guo and Li [19] use generalized moment estimation to empirically analyze the impact of scientific and technological financial investments on the manufacturing innovation efficiency. Based on the panel data of 82 countries from 2001 to 2018, Zhang and Chen [20] empirically test the impact of bank credit expansion on manufacturing upgrading from the perspective of R&D innovation. Finally, does this have a "spatial spillover" effect? According to financial geography emerging in the 1970s [21], financial centers have more obvious characteristics of spatial hierarchical distribution and location agglomeration, especially under the development of information technology, leading to the geographical agglomeration of financial institutions. There are similar conclusions among Chinese scholars, such as the center–outside theory, expectation theory, and so on. Wang [22] studies the spatial impact of financial innovation on the local economy, believing that the financial center is directly proportional to the level of urban construction, and that the construction of urban centers mostly relies on the construction of financial centers. In fact, agglomeration and spatial spillover also occur in manufacturing, but few scholars have studied them. The most remarkable feature of manufacturing intelligence upgrading itself is manufacturing agglomeration and its substitution and complementary effects on the labor force. Not only will the development of intelligent technology in a region promote the upgrading of the manufacturing industry, but so will factors such as labor transfer and

technology spillover among regions, as well as the effect of industries and enterprises on the factor structure of other regions [23].

In summary, many scholars from all over the world have paid attention to the impact of finance on manufacturing from the perspective of technological innovation [12,13]. However, on the one hand, scholars have not reached a consensus about whether financial innovation influences the intellectualized transformation and upgrading of the manufacturing industry. On the other hand, few studies take geographical factors into account and study the impact of financial innovation on manufacturing intelligence upgrading from the perspective of the spatial effect. Furthermore, the existing literature mostly focuses on the impact of the characteristics of finance on the real economy, which makes the research on the linkage mechanism between financial innovation and the intelligent upgrading of the manufacturing industry still insufficient. Therefore, we mainly study two aspects: first, according to literature analysis and theoretical research, we comb the mechanism of financial innovation affecting the intelligent transformation and upgrading of the manufacturing industry based on the typical characteristics of financial innovation and intelligent manufacturing in the digital era; second, we calculate the intelligent upgrading level of China's manufacturing industry and the efficiency of financial innovation from 2008 to 2019, then we use the spatial econometric model to empirically test the spatial effects, and further test the role of the mechanism.

Compared with prior studies, this study makes the following contributions to enriching this topic. First, this paper tests the impact on the intellectualized transformation and upgrading of the manufacturing industry from the perspective of financial innovation, which not only provides a new perspective, but also strengthens the positive research. Second, we construct a three-level index system, including basic investment, production application, and market efficiency, to evaluate the intelligence level of the manufacturing industry. According to the measurement results of the entropy weight method, this paper tests the spatial effect of financial innovation on the intellectualized transformation and upgrading of the manufacturing industry, which enriches the relevant research. Third, we explore the internal mechanism of the impact of financial innovation affecting the intelligent transformation of the manufacturing industry and test the intermediary effects of three mechanisms: consumption demand, capital allocation, and technological innovation. The study expands the research space of financial development and manufacturing upgrading.

The remainder of this paper proceeds as follows. Section 2 comprehensively reviews the financial innovation theory and puts forward three assumptions about financial innovation and the intelligent transformation and upgrading of the manufacturing industry. Section 3 describes our model, sample, and data selection. Section 4 presents the empirical test and result analyses. Section 5 provides conclusions and recommendations.

## 2. Theory and Hypotheses

### 2.1. Financial Innovation Theory

Financial innovation theory originates from Schumpeter's innovation theory [24], which mainly expounds the connotation and motivation of financial innovation. Scholars interpret the connotation of financial innovation from three perspectives: micro, meso, and macro. From the micro perspective, Tufano [25] divides financial innovation into product innovation and process innovation and puts forward that product innovation is the basic connotation of financial innovation, which has certain limitations. Based on the meso perspective, financial innovation is no longer limited to the innovation of tools and products but sets the perspective more broadly at the level of system and organization [26,27]. From a macro perspective, it is considered that financial innovation is a systematic, overall, and comprehensive financial reform, a financial industry reform under the new needs of economic development, specifically manifested in the combination of regional financial elements in a new way [25,28]. To sum up, according to the understanding of scholars, and providing theoretical support for the later measurement of the efficiency of financial innovation, this paper defines financial innovation as a new combination of factors formed

by the market, systems, and other forces based on products, technology, and other factors from a meso and macro perspective.

Financial innovation is the inevitable result of financial development [29]. The motivation to promote financial innovation includes technological progress [30,31], profit inducement [32], regulatory evasion [33], and institutional reform [34]. Among them, technological progress is one of the main driving forces. Hannan and McDowell [35] proposed that applying the invention of computers to the financial field would significantly promote the innovation of the financial industry. The emergence of the Internet has significantly promoted the speed and scale of financial innovation, and technological progress has driven price competition in financial activities [36]. The application of communication equipment and technology in the financial industry has greatly promoted financial innovation, reduced transaction costs, improved the matching efficiency between borrowers and lenders, and thus improved the efficiency of financial intermediaries [37]. With the enrichment and development of connotation, scholars mainly focus on the combination of financial innovation and the industrial economy and the real economy [6–8]. Enterprise development needs a lot of financial support and has the characteristics of being long-term and high-risk. Therefore, relying on external financing, such as financial institutions, has become an important source of enterprise innovation and transformation. However, the traditional financial support has asymmetric information and high financing cost problems, which leads to the difficulty and high cost of financing [38]. As it happens, digital technology and financial innovation is in continuous development, including the introduction of financial new products and the recombination of various financial elements, and has triggered a fundamental change in funds raising, service means, and credit preference, and has expanded financial coverage and service depth. All those radically solve the problem of asymmetric information between financial intuitions and manufacturing enterprises, which are faced with the severe challenges of intelligent transformation and upgrading [39]. The financing demand for intelligent transformation and upgrading has great uncertainty. With the support of financial innovation, financing services become transparent and smooth. Patrick [40] proposed that financial innovation affects the upgrading of the manufacturing industry through "supply guidance" and "demand induced". Supply guidance refers to the effective allocation of funds by the financial system according to the capital needs of different manufacturing industries, and demand-induced refers to the optimization of capital allocation due to the upgrading of the manufacturing industries. To this end, we propose the following hypothesis:

**Hypothesis 1.** *Financial innovation can effectively promote the intelligent transformational upgrading of the manufacturing industry.*

*2.2. Mechanism of Financial Innovation Affecting Intelligent Transformational Upgrading of the Manufacturing Industry*

The core function of financial innovation is to realize the effective allocation of funds, including financial instruments innovation, financial markets innovation, and institutions innovation. The intellectualized transformational upgrading process of the manufacturing industry is inseparable from the support of finance. Based on the theory, we consider that financial innovation may promote the intellectualized transformation and upgrading of the manufacturing industry through consumer demand, capital allocation, and technological innovation. The specific impact mechanism is shown in Figure 1.

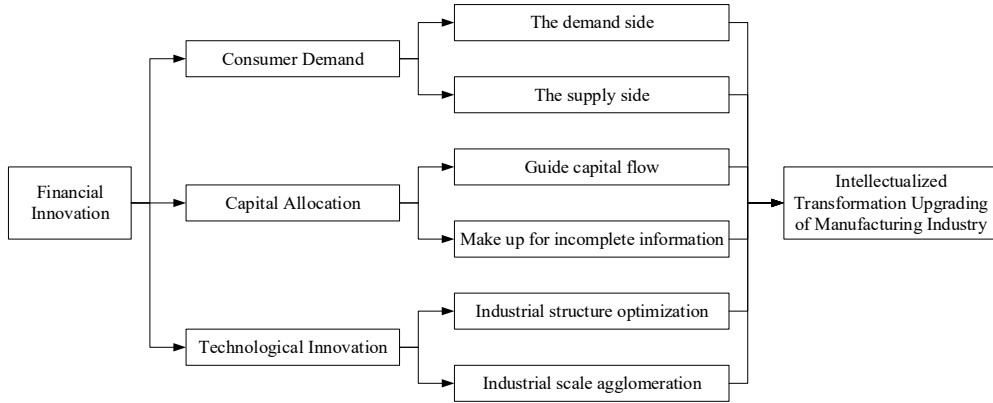

**Figure 1.** Mechanism of financial innovation affecting intelligent transformational upgrading of the manufacturing industry.

First, the consumer demand mechanism. Financial innovation promotes the intelligent transformation of the manufacturing industry from the demand side and supply side by triggering the change of consumer demand. From the demand side, financial derivatives are constantly enriched, and the financial market is constantly improved, providing residents with diversified and multilevel financial consumer products, such as insurance, funds, and financial products, as well as e-commerce consumption, phased shopping platforms, etc., which will affect consumer behavior. Specifically, by stimulating consumers' demand for more high value-added products and services, financial innovation will affect the total demand of consumers and the change of the consumption structure, and then form a traction on the supply side of the manufacturing industry. From the downstream to the upstream of the industrial chain, the change of the consumption demand will be integrated into the design, production, manufacturing, marketing, service, and other links of the manufacturing industrial chain, so as to promote the intelligent transformation of manufacturing industry [41]. From the supply side, financial institutions continue to innovate credit products and services, launching payment functions such as mobile payment and credit card payment. Compared with the traditional means of payment, this greatly improves the efficiency of consumer transactions and accelerates the transformation and circulation process between money, goods, and services [42]. This process directly affects relevant manufacturing industries from two aspects: the increase of total capital supply and the acceleration of the circulation speed, so as to provide financial support for the development of emerging industries and promote the optimization of the manufacturing structure.

Second, the capital allocation mechanism. To realize the optimal allocation of capital is the basic function of the financial system. The improvement of capital allocation efficiency refers to the flow of capital from industries with a low marginal rate of return to industries with a high marginal rate of return, and finally to achieve Pareto optimization [43]. In this process, the development level of the financial market is one of the important factors. First, the orientation and policy of financial innovation helps optimize the direction of capital allocation. By disclosing the investment return rate, risk assessment, financial status, and other information of manufacturing enterprises, it will guide the flow of funds to enterprises with higher profits and better development prospects [44]. In addition, the government attaches great importance to financial forces to help the intelligent transformation of the manufacturing industry, such as targeted RRR reduction and inclusive financial services. Driven from the supply side, it is easier for emerging transformational enterprises to obtain financing, which helps the manufacturing industry to realize intelligent upgrading. Second, through innovation and development, the financial market connects financial institutions with enterprises and between enterprises. By means of the information collection ability of financial intermediaries, it not only effectively reduces the capital mismatch between enterprise demand subjects and financial suppliers, but also provides rules and platforms for capital conversion, eliminates internal incomplete information, and provides sufficient

funds for the intellectualization of the traditional manufacturing industry. At the same time, a profit-driven capital flow can quickly adjust the existing stock structure, accelerate capital accumulation, and optimize the capital allocation structure. By driving the development of the advanced manufacturing industry and eliminating backward enterprises, it will have a positive impact on the intelligent upgrading of the manufacturing industry.

Third, the technological innovation mechanism. The fundamental driving force of the intelligent upgrading of the manufacturing industry is the integration of existing technology with intelligent technologies, such as the industrial Internet, big data, and cloud computing, which runs through all links such as design, production, and management. Technological innovation needs a large number of sustained and stable financial resources. First, on the basis of meeting the needs of different capital subjects, the development of financial intermediaries has effectively reduced the information asymmetry between the capital demander and the capital supplier. Diversified financing channels and financing methods, such as private capital and financial platforms, not only directly provide financial support for enterprises, but also accurately and efficiently guide the flow of funds into enterprises with a high rate of return, especially high-tech intelligent manufacturing enterprises [45]. All of these will promote the optimization of the intelligent manufacturing industry. Second, financial innovation itself will cause financial agglomeration and accelerate the intelligent manufacturing enterprises from the development stage to the growth stage [46]. At this stage, enterprises benefit from Jacob's external economy, which will enhance the sustainability of technological innovation. At the same time, according to the "polarization trickle-down effect", technological innovation will promote the expansion of the internal scale of the high-tech manufacturing industry. In addition, financial development will reduce information asymmetry among industries and reduce financial risks. It makes the upstream and downstream of the industry form a "point-to-surface" cluster development, so as to promote the agglomeration of intelligent manufacturing.

**Hypothesis 2.** *Financial innovation may improve the intelligent upgrading level of the manufacturing industry through consumer demand, capital allocation, and technological innovation.*

### 2.3. Spatial Spillover Effects Related to Financial Innovation and Intelligent Transformation of the Manufacturing Industry

In the era of the digital economy, financial innovation and the intelligent transformation of the manufacturing industry are fully integrated with digital information technology, showing new characteristics such as breaking through geographical distance constraints and capital financing limitations, which significantly enhances the depth and breadth of the depth and width of financial innovation activities and the diffusion of factor resources among regions. Therefore, when examining the impact of financial innovation on the intelligent transformation of the manufacturing industry in the region and surrounding areas, spatial-related factors should be fully considered [47]. Specifically, financial transactions, financial information, financial elements, and knowledge and technology elements have natural liquidity and spillover. Driven by informatization, the diffusion cost is reduced, and its speed is accelerated, which makes the elements of financial innovation and the upgrading of manufacturing intelligence show strong spatial spillover [48]. Ignoring the spillover effect of financial innovation may affect the effectiveness of the evaluation results of the manufacturing transformation. Because of this, we try to explain the spatial spillover effect of financial innovation on the intelligent transformation and upgrading of the manufacturing industry from the "siphon effect" and "trickle-down effect" [49–51].

First of all, financial development is a cumulative but discontinuous process which usually originates from areas with superior conditions and gradually expands to surrounding areas [52,53]. At the initial stage, the regions with a high efficiency of financial innovation and a high resource endowment for intelligent transformation and upgrading of the manufacturing industry take the lead in the development [52,54]. The demand for knowledge, technology, capital, labor, and other factors in this region is stronger [55]. According to the siphon effect, a higher marginal rate of return of factors is triggered,

attracting the cross-regional inflow of factors, and the production factors related to the development of the manufacturing industry begins to concentrate. In the end, a few areas will be developed into high concentration areas of intelligent manufacturing [49]. Secondly, with the development of financial innovation, due to the development needs of the manufacturing industry and the profit-seeking nature of capital itself, the manufacturing industry begins to develop to the surrounding areas [50]. According to the "trickle-down effect", the priority areas promote the development of the surrounding areas through demonstration and imitation or knowledge and technology spillover [26,51], resulting in the spatial spillover effect. The spatial spillover effect here means that the development of financial innovation accelerates the development and competition of the manufacturing industry in neighboring cities. On the one hand, financial innovation leads to manufacturing agglomeration, accelerates the liquidity of production factors, improves the efficiency of capital use, and maintains industrial competitiveness by improving technology [26]; on the other hand, transferring scientific research talents outward strengthens exchanges among the surrounding areas [51]. For this reason, financial innovation can not only weaken the adverse impact of information asymmetry between the capital supply and demand in the region to a certain extent, but also bring about intelligent transformation and the upgrading of the manufacturing industry in neighboring regions. The application of digital technology and other scientific and technological means to the financial field provides support for long-distance technological innovation. It has the advantages of "low cost, fast speed and high degree of informatization", which gradually reduces the impact of geographical distance on the development of the manufacturing industry, reduces the friction coefficient between spaces, and promotes the spatial spillover of the intelligent manufacturing industry [39]. Based on this, we propose the following hypothesis:

**Hypothesis 3.** *Financial innovation can effectively promote the intelligent transformation and upgrading of the manufacturing industry in the region. Apart from this, it produces spatial spillovers to the surrounding areas.*

### 3. Methods and Data

#### 3.1. Sample Selection

The panel data of 30 provinces from 2008 to 2019 in China are our research samples. We mainly study the impact of financial innovation on the intelligent transformation and upgrading of the manufacturing industry. The research objects selection and research span are very important. Due to the serious lack of key variable data, we selected 30 provinces as the main research objects, except Tibet, Hong Kong, Macao, and Taiwan. As for the research span, until 2008, through 30 years of reform and opening up, China's manufacturing industry has basically maintained a growth rate of 12–14%. However, as the result of the depreciation of the US dollar, the rise of raw material prices, energy conservation, and emission reduction, etc., the development of the manufacturing industry has been impacted and urgently needs transformational upgrading. At the same time, as information technology develops, opportunities for the intelligent transformation of the manufacturing industry have been sought. Data in 2019 is the latest according to official statistics, which is why we chose data from 2008 as a start to do the deep analysis.

#### 3.2. Model Construction and Variable Measurement

Space weight matrix setting. The premise of spatial measurement is to measure the spatial distance between regions, and the key is to set the spatial weight matrix. The commonly used weight matrices include adjacent distance, geographical distance, economic distance, cultural distance, etc. Considering that the economic development level of regional units will have an interactive impact between spatial units, we choose the economic distance weight matrix based on the research of Liu and Ma [56], which refers to the

geographic distance weight matrix modified by the diagonal matrix, constructed by using the economic development level index of the research object. The formula is as follows:

$$W_{ij} = \begin{cases} \frac{1}{\left|gdp_i - gdp_j\right|}, & i \neq j \\ 0, & i \neq j \end{cases} \tag{1}$$

In Formula (1), $W_{ij}$ is the element in the matrix, which represents the weight of the economic distance between regions i and j, while $gdp_i$, $gdp_j$ represent the average gdp of region iand j in the statistical period, respectively.

Construction of the spatial econometric model. This paper mainly studies the impact of financial innovation on the manufacturing industry, but it will affect the intelligent transformation of the manufacturing industry in this region as well as the surrounding and adjacent areas. Compared with the ordinary panel regression model, the spatial panel econometric model fully considers the influence of spatial factors, which has a higher fitting degree, including the spatial lag model (SAR), spatial error model (SEM), and spatial Dobbin model (SDM) [57]. Among them, the SAR model indicates that there is an interdependent spatial interaction between the manufacturing intelligent transformation and the upgrading levels of adjacent provinces. The SEM model shows that the spatial dependence is reflected by the error term. The SDM model shows that the financial innovation of both this province and the surrounding provinces will affect the level of manufacturing intelligent transformation and upgrading in this province. That means there are interdependent spatial externalities that exist. The expression is as follows:

$$LnMfg_{it} = \rho \times W \times LnMfg_{it} + \sum_{j=1}^{N} W \times LnFin_{ijt} \times \theta + \alpha LnFin_{ijt} + \mu_i + \xi_i + \varepsilon_{it} \tag{2}$$

$$\varepsilon_{it} = \lambda \times W \times \varepsilon_t + \nu_{it} \tag{3}$$

where $LnMfg_{it}$ is the dependent variable, which is the intelligent upgrading level of the manufacturing industry, $\rho \times W \times LnMfg_{it}$ is the spatial lag term of the dependent variable,$\rho$ is the spatial autocorrelation coefficient, which measures the impact of adjacent areas on $LnMfg_{it}$, Wrepresents the spatial weight matrix, $LnFin_{ijt}$ is the explanatory variable, $\sum_{j=1}^{N} W \times LnFin_{ijt}$ represents the spatial lag term of the explanatory variable, $\theta$ measures the impact of adjacent areas on $LnFin_{ijt}$; $\mu_i$, $\xi_i$ stands for the space effect and time effect, respectively; $\varepsilon_{it}$ is the spatial error perturbation term and $\lambda$ represents the spatial dependence of $LnMfg_{it}$. Specifically, if $\lambda = 0$ and $\theta = 0$, it belongs to the SAR model; if $\rho = 0$ and $\theta = 0$, it belongs to the SEM model; if $\lambda = 0$, it is the SDM model.

*3.3. Variable Description*

Dependent variable. The intelligent transformation and upgrading of the manufacturing industry (Lnmfg). The intelligent transformation and upgrading of the manufacturing industry is based on the deep integration of the industrial Internet, big data, metacomputing, and other architectures in the advanced manufacturing industry. It intrinsically requests the comprehensive intelligence of the whole chain of the manufacturing industry, such as R&D, production, supply, sales, and service, far more than the intelligence of "manufacturing" [58]. While constructing its index system, this paper considers not only the connotation of intelligent development in the era of the digital economy, but also the actual situation of China's manufacturing development at this stage. On the basis of the research of scholars such as Li.et al. [59] and Yu.et al. [60], the intelligent transformation of the manufacturing industry includes all links of activities, such as design, production, and management. Specifically, the factor input is the basis of the intelligent transformation of the manufacturing industry, including R&D, funds, intelligent equipment, Internet infrastructure, and personnel investment. The development and application of software technology

is the key to the sustainable transformation of the manufacturing industry, which is mainly reflected in equipment intelligence, management intelligence, product intelligence, and service intelligence. The ultimate goal is to achieve better economic and social benefits, such as environmental improvement and energy intensity. Therefore, as shown in Table 1, it is constructed by three elements, the basic input level, the production application level, and the market benefit level, with 10 secondary indicators and 15 tertiary indicators.

The entropy weight method is used to calculate the intelligent upgrading level of the manufacturing industry. The higher the value, the higher the level of the intelligent transformation and upgrading.

Independent variable. Financial innovation ($Lnfin$) refers to the internal innovation of financial institutions. According to this understanding, financial institutions are often regarded as a production unit to measure the creation of their financial products based on their input and output [26,27]. The existing evaluation methods for financial innovation mainly include stochastic boundary function analysis (SFA) [61,62], the free step-by-step method (DFA), and data envelopment analysis (DEA) [61,63,64]. Among them, because of the DEA model's advantages in dealing with multiple inputs and outputs, it has been widely used. Moreover, financial innovation is more reflected in the optimal allocation of production factors. Efficiency can just reflect the utilization capacity of production factors, and it is more suitable for analyzing the service characteristics of the financial industry with more input and more output [65]. Llewellym [66] proposed that the improvement degree of efficiency can reflect innovation. Therefore, this paper carries out research practice from the perspective of financial innovation efficiency and constructs an input–output model through the DEA model to measure the financial innovation of each province [67]. Specifically, the input variable draws on the experience of Fernández et al [68] to select the capital stock and number of employees in the financial industry, while the output variable selects the added value of the financial industry. The calculation of capital stock is the same as that of Chen [69], taking 2008 as the base period, and then using the perpetual inventory method. The formula is as follows:

$$K_{i(t+1)} = \left(1 - \sigma_{it}K_{it}\right) + I_{i(t+1)} \tag{4}$$

In the formula, $K_{i(t+1)}$, $K_{it}$ respectively represent the financial capital stock of province i in year t + 1 and year t, $\sigma_{it}$ refers to the capital depreciation rate in year t of province i, and $I_{i(t+1)}$ refers to the new fixed asset investment in the financial industry in the year t + 1 of province *i*. The initial capital stock is calculated by dividing the new fixed assets of the financial industry in the base period by 10%, and the investment in new fixed assets of the financial industry is reduced.

Control variables. In fact, the intelligent upgrading level of the manufacturing industry is affected by many responsible factors. Various influencing factors have been studied by scholars, including the urbanization level, human capital level, industrial upgrading level, infrastructure construction, and government subsidies [70–73]. Therefore, according to the previous literature, five variables are selected as control variables. Specifically, the urbanization level (Lncity) is expressed by the ratio of the urban permanent population to the total regional permanent population [22,70]; the level of human capital (Lnedu) is measured by the number of full-time teachers in colleges and universities [71]; the level of industrial upgrading (Lnins) is characterized by the proportion of tertiary industry and secondary industry in GDP, respectively [72]; infrastructure construction (Lntrans) is expressed by the ratio of total freight volume to GDP [73]; government subsidy (Lngov) is measured by the ratio of science and technology expenditure to the general public expenditure in the government budget [17].

**Table 1.** Evaluation index system of intelligent transformation and upgrading of manufacturing industry.

| Main Index | Primary Index | Secondary Index | Tertiary Indicators | Measure Index |
|---|---|---|---|---|
| Intelligent transformation and upgrading of manufacturing industry | Foundation input layer | R&D investment | R&D investment in high-tech manufacturing industry | R&D funds for high-tech manufacturing/R&D funds for industrial enterprises |
| | | Intelligent device input | Manufacturing fixed assets investment | Fixed asset investment in telecommunications/fixed asset investment in manufacturing |
| | | | Investment in fixed assets of information service industry | Investment in fixed assets of information technology service industry |
| | | Internet based investment | Internet coverage | Optical cable line length/provincial area |
| | | | Internet penetration | Number of Internet users/population in province |
| | | Personnel input | Number of employees in high-tech manufacturing industry | Number of employees in high-tech manufacturing industry/number of employees |
| | | | Number of personnel in information transmission, software and information technology services | Number of personnel/employees in information transmission, software and information technology services |
| | Production application layer | Equipment intelligence | Software development and service | Software business income/manufacturing business income |
| | | Management intelligence | Data processing and operation | Sum of software product revenue and embedded system software revenue/manufacturing owner's business revenue |
| | | Product intelligence | Industrialization degree of Intelligent Technology | Output value of new products in high-tech manufacturing industry/business income of manufacturing owners |
| | | | | Effective invention patents of high-tech manufacturing industry/effective invention patents of industrial enterprises |
| | | Service intelligence | Information service | Information technology service income/manufacturing owner's business income |
| | Market benefit layer | Economic performance | Smart device market profit | Total profits of high-tech manufacturing |
| | | | Smart device market efficiency | Business income/number of employees of high-tech manufacturing owners |
| | | Social results | Environmental improvement | Manufacturing wastewater discharge/manufacturing added value |
| | | | | Manufacturing emissions/manufacturing value added |
| | | | | Industrial solid waste emissions/manufacturing value added |
| | | | Energy intensity | Manufacturing energy consumption/manufacturing value added |

Table 2 shows the descriptive statistics of the relevant variables. It mainly comes from the China Statistical Yearbook, the China High-Tech Industry Statistical Yearbook, the China Financial Statistical Yearbook, the China Energy Statistical Yearbook, the China Listed Company Statistical Yearbook, and the Tai'an database. Some missing data are supplemented by the provincial statistical yearbook, the interpolation method, or the mean substitution method.

**Table 2.** Descriptive statistical analysis of variables.

| Variable | Describe | Sample Size | Mean Value | Standard Deviation | Minimum Value | Maximum |
|---|---|---|---|---|---|---|
| Dependent variable | Intelligent transformation and upgrading level of manufacturing industry | 360 | 0.135 | 0.084 | 0.025 | 0.605 |
| Independent variable | Financial innovation | 360 | 0.547 | 0.214 | 0.199 | 1.000 |
| Control variables | Urbanization level | 360 | 0.558 | 0.130 | 0.291 | 0.896 |
| | Human capital level | 360 | 9.817 | 1.171 | 6.991 | 13.829 |
| | Industrial upgrading level | 360 | 1.000 | 0.412 | 0.010 | 2.016 |
| | Infrastructure construction | 360 | 0.050 | 0.014 | 0.022 | 0.104 |
| | Government grants | 360 | 0.004 | 0.003 | 0.001 | 0.034 |

## 4. Results and Discussion

### 4.1. Analysis of Measurement Results of Intelligent Transformation and Upgrading Level of the Manufacturing Industry

By observing the changing trend of the intelligent transformation and upgrading of the manufacturing industry, the four typical years of 2008, 2012, 2016, and 2019 are selected to analyze the spatial evolution characteristics of China's manufacturing industry.

Figure 2 is a schematic diagram which uses the natural breakpoint method to divide the one-year manufacturing intelligence upgrading level into four levels. First of all, in terms of time evolution, the intelligence level of the manufacturing industry has been improved in most regions since 2008. Each province changes stably, besides that of Liaoning, Shaanxi, and other provinces, which have always maintained a high level. Second, in terms of spatial distribution, the intelligent upgrading level of the manufacturing industry in all provinces generally presents an unbalanced state. The gap between the eastern region is relatively small, and the central and western regions of China are relatively large, which are characterized by being high in the east and low in the west. The gap of the level between the adjacent provinces is similar, showing obvious spatial agglomeration characteristics. Third, in terms of spatial evolution, the intelligent upgrading level of the manufacturing industry shows a dynamic increasing trend from the western interior to the eastern coastal areas, with an obvious economic trend. There is a clustering phenomenon in Shandong, Anhui, Hubei, Zhejiang, Fujian, and other provinces in the east, and this gradually shows a regional driving effect. It can be concluded that the development of the intelligent upgrading level of the manufacturing industry in each province is uneven, which is the same as the research conclusion of Han et al. [74].

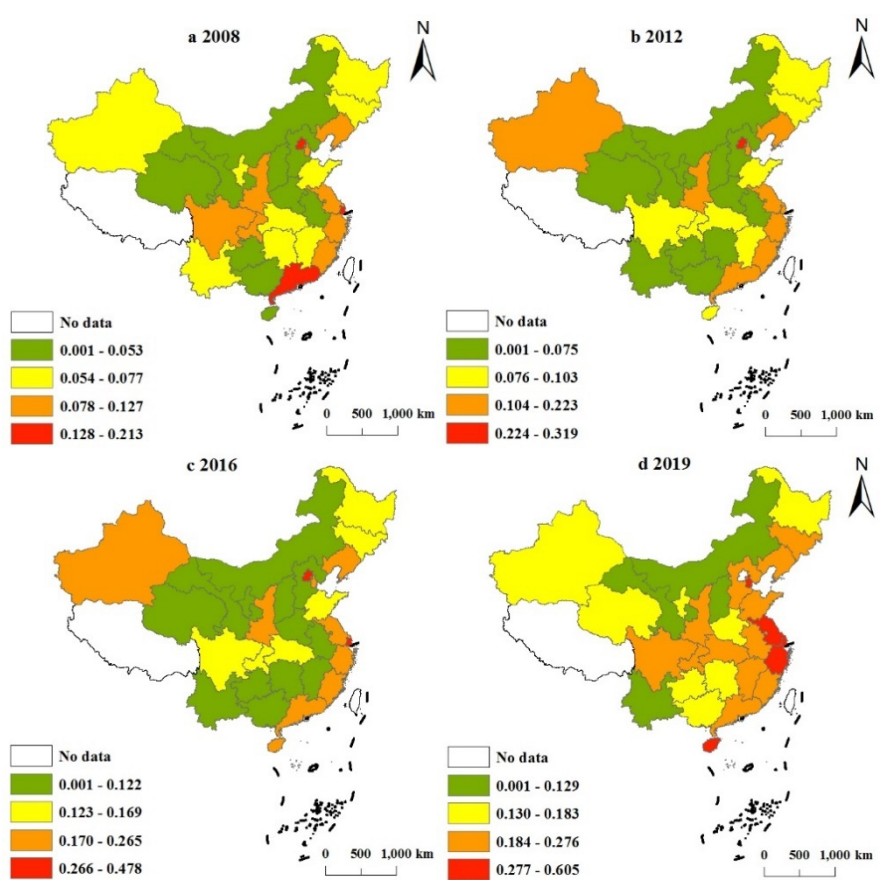

**Figure 2.** Temporal and spatial characteristics of intelligent transformation and upgrading of China's manufacturing industry in 2008, 2013, 2014, and 2019.

### 4.2. Empirical Test of Spatial Effect

#### 4.2.1. Spatial Correlation Test

We firstly perform the global spatial autocorrelation test. The premise of using the spatial econometric model is to clarify the correlation relationship between spatial variables [50,57]. The most commonly used spatial autocorrelation test is Moran's I test. The larger the value of Moran's I, the higher the degree of spatial autocorrelation. More than 0 indicates positive autocorrelation between high values, low values are the same, and less than 0 indicates the existence of negative autocorrelation. As shown in Table 3, the Moran's I is positive through the spatial correlation test under the weight matrix dimension of geographical distance and economic distance. Except that the geographical distance weight matrix failed to pass the significance test in 2014, the other *p*-value values are less than 0.1. Therefore, under the 90% confidence, it can be considered that the intellectualization of China's provincial manufacturing industry is not isolated and randomly distributed from 2008 to 2019, and there is a spatial spillover effect.

**Table 3.** Global Moran index test results of intelligent transformation and upgrading of China's provincial manufacturing industry from 2008 to 2019.

| Year | Geographical Distance | | | | | Economic Distance | | | | |
|------|------|------|------|------|------|------|------|------|------|------|
| | I | E (I) | sd (I) | z | *p*-Value | I | E (I) | sd (I) | z | *p*-Value |
| 2008 | 0.160 ** | −0.034 | 0.091 | 2.151 | 0.031 | 0.246 *** | −0.034 | 0.082 | 3.422 | 0.001 |
| 2009 | 0.185 ** | −0.034 | 0.090 | 2.444 | 0.015 | 0.223 *** | −0.034 | 0.081 | 3.174 | 0.002 |
| 2010 | 0.156 ** | −0.034 | 0.087 | 2.180 | 0.029 | 0.240 *** | −0.034 | 0.079 | 3.467 | 0.001 |
| 2011 | 0.133 * | −0.034 | 0.088 | 1.907 | 0.057 | 0.266 *** | −0.034 | 0.079 | 3.780 | 0.000 |
| 2012 | 0.154 ** | −0.034 | 0.089 | 2.109 | 0.035 | 0.266 *** | −0.034 | 0.081 | 3.717 | 0.000 |
| 2013 | 0.144 ** | −0.034 | 0.089 | 2.020 | 0.043 | 0.261 *** | −0.034 | 0.080 | 3.688 | 0.000 |
| 2014 | 0.112 | −0.034 | 0.090 | 1.630 | 0.103 | 0.199 *** | −0.034 | 0.082 | 2.858 | 0.004 |
| 2015 | 0.119 * | −0.034 | 0.089 | 1.732 | 0.083 | 0.244 *** | −0.034 | 0.080 | 3.467 | 0.001 |
| 2016 | 0.115 * | −0.034 | 0.087 | 1.717 | 0.086 | 0.260 *** | −0.034 | 0.079 | 3.729 | 0.000 |
| 2017 | 0.144 ** | −0.034 | 0.085 | 2.088 | 0.037 | 0.251 *** | −0.034 | 0.077 | 3.696 | 0.000 |
| 2018 | 0.116 * | −0.034 | 0.081 | 1.855 | 0.064 | 0.211 *** | −0.034 | 0.074 | 3.327 | 0.001 |
| 2019 | 0.196 *** | −0.034 | 0.085 | 2.713 | 0.007 | 0.197 *** | −0.034 | 0.077 | 3.013 | 0.003 |

Note: ***, **, and * denote coefficients significant at 1%, 5%, and 10% statistical levels, respectively.

Secondly, we test the local spatial correlation in order to further determine the impact of the financial innovation level of each province on the intelligent transformation and upgrading of the manufacturing industry. For the systematic and scientific study, two value points (2008, 2019) are selected for analysis. Figure 3 is a bivariate Moran scatter diagram of two key variables, financial innovation and manufacturing intelligence upgrading level. It can be seen that Moran's I values in 2008 and 2019 were 0.732 and 0.562, respectively, both of which were significantly positive. Table 4 shows the specific distribution areas of each province.

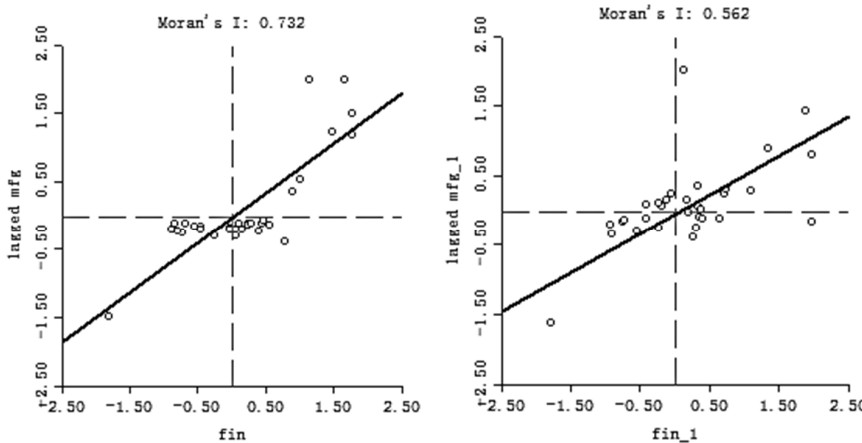

**Figure 3.** Bivariate Moran scatter diagram (2008/2019).

**Table 4.** Distribution area of bivariate Moran scatter diagram.

| Quadrant | 2008 | 2019 |
|----------|------|------|
| First quadrant High concentration | Beijing Tianjin Shanghai Fujian Guangdong Qinghai Jiangsu | Beijing Tianjin Shanghai Fujian Guangdong Qinghai Jiangsu Zhejiang Hubei Chongqing Sichuan Guangxi |
| Beta Quadrant Low high concentration | Nothing | Shandong Jiangxi Hainan Anhui Guizhou |
| Third quadrant Low concentration | Hebei Shanxi Liaoning Jilin Henan Ningxia Gansu Xinjiang Heilongjiang Inner Mongolia | Hebei Shanxi Liaoning Jilin Heilongjiang Hunan Inner Mongolia |
| Delta Quadrant High and low agglomeration | Zhejiang Anhui Jiangxi Shandong Hubei Guangxi Hainan Chongqing Sichuan Guizhou Yunnan Shaanxi | Henan Yunnan Shaanxi Gansu Ningxia Xinjiang |

From Figure 3, the high-high type is located in the first quadrant. These provinces with a high efficiency of financial innovation are clustered around other provinces with a high level of intelligent transformation and upgrading of the manufacturing industry, indicating that local finance supports the development of the manufacturing industry. The provinces in the second quadrant belong to the low-high type, indicating that the provinces with a low efficiency of financial innovation are surrounded by provinces with a high level of intelligent transformation of the manufacturing industry. The third quadrant is low-low type. The financial innovation efficiency of provinces in this quadrant and the intelligent transformation and upgrading level of the manufacturing industry in the surrounding provinces are low, which means that the low financial innovation efficiency may be one of the reasons that hinder the intelligent transformation of the local manufacturing industry. Those in the fourth quadrant are high-low, which may mean that the high level of financial innovation in these provinces does not provide effective support for the manufacturing industry.

In Table 4, compared with 2008, the high-high type in 2019 increased by five provinces compared with 2008, and all of them came from the high-low agglomeration type. This shows, with the passage of time, the improvement of the financial innovation efficiency in five provinces, including Zhejiang, Hubei, and Sichuan, has significantly enhanced the support effect for the intelligent transformation and upgrading of the manufacturing industry. The transfer of the high-low cluster provinces to the low-high cluster provinces include five provinces, such as Shandong, Jiangxi, and Hainan. It shows that these provinces and their adjacent regions have realized the intelligent transformation and development of the manufacturing industry in recent years. However, it is not due to the improvement of the financial innovation efficiency, but other factors that make up for the lack of local financial development. Taking Shandong Province as an example, as the inherent advantage of Shandong's economy, although the financial development level of this province is low, the intelligence level of the manufacturing industry has been significantly improved with the help of policy guidance and industrial foundation through adhering to the deep integration of digital technology in the manufacturing industry. Ningxia, Gansu, and Xinjiang have been struck off the list from the low-low provinces, which indicates that the development level of financial innovation in these provinces has been significantly improved in recent years.

### 4.2.2. Spatial Econometric Analysis

Table 5 shows the test results of the key link of model identification in the construction of the spatial econometric model [15,57]. Firstly, the LM and robust LM test methods are used to determine whether the spatial panel model adopts lag model or error model. According to the measurement results in Table 5, the $p$-values of the LM spatial lag, robust LM spatial lag, LM spatial error, and robust LM spatial error are less than 0.05. The statistics have passed the significance test at the level of 5%, indicating that the impact of financial innovation on the intelligent upgrading of the manufacturing industry has the form of a spatial error and a spatial lag at the same time. Secondly, judging whether the SDM model can degenerate into the SAR and SEM by observing the coefficient significance of Wald and LR tests. The statistics of the Wald spatial lag, Wald spatial error, LR spatial lag, and LR spatial error are significant at the level of 1%, thus rejecting the original hypothesis that the optimal model is the panel SDM of financial innovation efficiency on the intelligent upgrading of the manufacturing industry. Finally, using the Hausman test to judge whether to choose the fixed effect or the random effect. The value of the Hausman test is 31.21 and the $p$-value is 0.0031. This indicates that the fixed-effect model is more suitable for estimation. To sum up, the fixed-effect SDM model is selected as the best spatial econometric model [75].

**Table 5.** Identification and inspection results of spatial metrology model.

| Statistic | Number | *p*-Value | Statistic | Number | *p*-Value |
|---|---|---|---|---|---|
| LM—spatial lag | 54.663 *** | 0.000 | Wald—spatial lag | 17.730 *** | 0.007 |
| Robust LM—spatial lag | 6.380 ** | 0.012 | Wald—spatial error | 30.500 *** | 0.000 |
| LM—spatial error | 224.498 *** | 0.000 | LR—spatial lag | 40.050 *** | 0.000 |
| Robust LM—spatial error | 176.215 *** | 0.000 | LR—spatial error | 46.650 *** | 0.000 |
| LR test (spatial fixed effect) | 54.940 *** | 0.000 | LR test (time fixed effect) | 414.370 *** | 0.000 |
| Hausman test | 31.210 *** | 0.003 | | | |

Note: ***, and ** denote coefficients significant at 1%, and 5% statistical levels, respectively.

Table 6 shows the spatial Dobbin model of the time fixed effect, and the following conclusions are drawn: First, the SDM model with the time fixed effect is the optimal model. The $R^2$ is 0.717 and the sigma2 is 0.061, indicating a high degree of fitting. Most of the explanatory variables and the spatial lag terms pass the significance test, and the overall effect is good. Second, under the economic distance weight matrix, the coefficient of Lnfin is 0.085, which has a 10% significance level. This shows that the development of financial innovation can promote the intelligent transformation of the manufacturing industry in this region. These results are supported by studies in [18,76]. The significantly positive coefficient of the spatial lag term (W × Lnfin) indicates that the efficiency of financial innovation in this region can promote the development of intelligent manufacturing in neighboring provinces and regions. That means there is a positive spatial spillover effect which is consistent with the results of the spatial correlation test, and hypothesis H1 is proved.

**Table 6.** SDM estimation results of time fixed effect.

| Variables | Main | Wx | Spatial | Variance |
|---|---|---|---|---|
| Lnfin | 0.085 * (0.06) | 0.095 * (0.03) | | |
| Lncity | 0.714 *** (0.00) | 1.468 *** (0.00) | | |
| Lnedu | 1.386 *** (0.00) | −1.963 ** (0.02) | | |
| Lnins | −0.118 *** (0.00) | 0.024 (0.78) | | |
| Lntrans | −0.240 *** (0.00) | −1.041 *** (0.00) | | |
| Lngov | 0.007 (0.86) | −0.036 (0.74) | | |
| rho | | | 0.280 *** (0.01) | |
| sigma2_e | | | | 0.061 *** (0.00) |
| Observations | 360 | 360 | 360 | 360 |
| R-squared | 0.717 | 0.717 | 0.717 | 0.717 |
| Number of code | 30 | 30 | 30 | 30 |

Note: ***, **, and * denote coefficients significant at 1%, 5%, and 10% statistical levels, respectively. The p value is in brackets.

### 4.2.3. Endogenous Analysis

Table 7 shows the results of the endogenous test in order to avoid the deviation of the empirical results caused by missing variables and reverse causality. Learning from the practice of Feng and Zhang [77], the instrumental variable is constructed by the product of the lagging first-order financial innovation efficiency $Lnfin_{it-1}$ and its first-order difference $\triangle Lnfin_{it-1}$ in time to estimate the instrumental variable. Meanwhile, the estimation

results of the instrumental variable method are compared with those of the OLS regression and spatial econometric regression. It can be seen that the estimated coefficients of the core explanatory variable Lnfin is significantly positive, and the symbols of the other variables are also consistent. This shows that, after considering the potential endogenous problems, the efficiency of financial innovation still significantly promotes the upgrading of manufacturing intelligence, and again verifies the robustness of the result of Hypothesis H1.

**Table 7.** Endogeneity test.

| Variable | Lnfin | Lncity | Lnedu | Lnins | Lntrans | Lngov | $R^2$ | N |
|---|---|---|---|---|---|---|---|---|
| OLS model | 0.016 ** (0.04) | 0.729 *** (0.00) | 1.762 *** (0.00) | −0.051 * (0.09) | −0.296 *** (0.00) | 0.165 *** (0.00) | 0.671 | 360 |
| Instrumental variable method | 0.361 *** (0.00) | 1.021 *** (0.00) | 1.402 *** (0.00) | −0.023 (0.47) | −0.296 *** (0.00) | 0.299 *** (0.00) | 0.613 | 330 |

Note: ***, **, and * denote coefficients significant at 1%, 5%, and 10% statistical levels, respectively. The $p$ value is in brackets.

### 4.2.4. Spatial Spillover Effects

Table 8 shows the results of the spatial spillover effect test, which can further reveal the direct and indirect impact of financial innovation on the intelligent upgrading of the manufacturing industry [57]. The point estimate used in the SDM model cannot represent the marginal effect on the intelligent transformation and upgrading of the manufacturing industry, and, considering that SDM contains both independent variables and dependent variables, the spatial influence cannot be accurately reflected. Therefore, on this basis, the direct and indirect effects of the financial innovation efficiency on the manufacturing industry are calculated by the partial differential method [57]. The direct effect and indirect effect represent the impact of the regional innovation efficiency on the intelligent upgrading of the manufacturing industry in local and adjacent areas, respectively. It shows that the direct effect is 0.083, which is significant at the level of 10%, and indicates that the financial innovation efficiency can promote the intelligent upgrading of local manufacturing industry, while the corresponding indirect effect is 0.058. Although nonsignificant, it also proves that the financial innovation efficiency plays a positive role in promoting the surrounding areas. These results also confirm a study finding on the spatial spillover effect of manufacturing industry in India [78]. Moreover, the intraregional spillover effect is greater than that in inter-regions, and hypothesis H3 is confirmed.

**Table 8.** SDM spatial effect decomposition of time fixed effect.

| Variable | (1) Direct Effect | (2) Indirect Effect | (3) Total Effect |
|---|---|---|---|
| lnfin | 0.083 * (0.08) | 0.058 (0.50) | 0.141 * (0.09) |
| lncity | 0.649 *** (0.00) | 1.035 *** (0.00) | 1.684 *** (0.00) |
| lnedu | 1.526 *** (0.00) | −1.934 *** (0.00) | −0.408 (0.50) |
| lnins | −0.119 *** (0.01) | 0.051 (0.50) | −0.068 (0.28) |
| lntrans | −0.193 *** (0.00) | −0.812 *** (0.00) | −1.005 *** (0.00) |
| lngov | 0.009 (0.81) | −0.029 (0.75) | −0.021 (0.83) |
| Observations | 360 | 360 | 360 |
| R-squared | 0.717 | 0.717 | 0.717 |
| Number of code | 30 | 30 | 30 |

Note: ***, and * denote coefficients significant at 1%, and 10% statistical levels, respectively. The p value is in brackets.

*4.3. Mechanism Analysis*

Based on the theoretical analysis and the empirical analysis above, this paper further discusses the influence mechanism. Taking consumer demand, capital allocation, and technological innovation as the starting point, and setting up the SDM model, which is based on the time fixed effect, the analysis is as follows:

$$\text{medium}_{it} = \rho_0 + \rho_1 \text{treat}_{it} \times \text{time}_{it} + \rho_2 \text{control}_{it} + \gamma_t + \mu_i + \varepsilon_{i,t} \tag{5}$$

$$\text{LnMfg}_{it} = \rho \times W \times \text{LnMfg}_{it} + \rho_1 \times W \times \text{LnMfg}_{it} \times \text{medium}_{it} + \sum_{j=1}^{N} W \times LnFin_{ijt} \times \theta +$$
$$\sum_{j=1}^{N} W \times LnFin_{ijt} \times \text{medium}_{it} \times \theta_1 + \alpha LnFin_{ijt} + \beta \text{control}_{it} + \mu_i + \xi_i \tag{6}$$

In the formula, the control variable $\text{control}_{it}$ is consistent with Equation (2), and the study analyzes the impact of financial innovation on the intelligent upgrading of the manufacturing industry through the interaction term between the intermediary mechanism and financial innovation (Medium × Lnfin). Medium includes the consumption demand (Lnsum), capital allocation (Lncap), and technological innovation mechanism (Lnino). Consumer demand is expressed by the ratio of total retail sales of social consumer goods to GDP [42]. Technological innovation is measured by the number of invention patents authorized per 10,000 people [44]. Based on the research of Wurgler [79], capital allocation is represented by the reciprocal of capital price distortion. The calculation steps are as follows:

$$Y_{it} = AK_{it}^{\alpha_{Ki}} L_{it}^{1-\alpha_{Ki}} \tag{7}$$

$$Ln(Y_{it}/L_{it}) + LnA + \alpha_{Ki}Ln(Y_{it}/L_{it}) + \omega_{it} \tag{8}$$

$$\gamma_{Ki} = (K_i/K)/(s_i\alpha K_i/\alpha_K) \tag{9}$$

$$\tau_{K_i} = 1/\gamma_{Ki} - 1 \tag{10}$$

Firstly, we set the C-D production function Formula (7) and take the logarithm on both sides to sort out Formula (8), where $Y$ represents the actual GDP of each province, which is obtained by adjusting based on 2008; $K$ refers to the actual capital stock, which is calculated by using the perpetual inventory method based on 2008; L is the average annual employment of each province. Secondly, we calculate the elasticity of capital output $\alpha$ into Formula (9), where $s_i = y_i/Y$ represents the output $y_i$ of region $i$ share of output y of the whole economy; $\alpha_K$ represents the weighted capital contribution value; $K_i/K$ represents the actual proportion of the capital used by region $i$ in the total capital; $s_i\alpha K_i/\alpha_K$ refers to the theoretical proportion of regional $i$ use capital when capital is effectively allocated; the ratio $\gamma_{Ki}$ of the two reflects the deviation between the amount of capital actually used and the effective allocation. Finally, Equation (10) calculates the capital allocation index $\tau_{K_i}$.

Table 9 shows the mechanism regression results, which respectively represent the intermediary effects of the three mechanisms: consumption demand, capital allocation, and technological innovation, then decomposes the direct effect, indirect effect, and total effect of each spatial model [57]. We found, firstly, for different mechanisms, the regression coefficients of financial innovation efficiency Lnfin are positive as well as significant at the level of 1%, which indicates that the financial innovation efficiency has a significant positive spillover effect on manufacturing intelligence upgrading, which is consistent with H1 and H3. Second, the direct effect interaction items of the three mechanisms and financial innovation are 0.225, 0.131, and 0.063, respectively. They are all significant at the level of 5%, which confirms hypothesis H2, and means that the efficiency of financial innovation will affect the intelligent upgrading of the manufacturing industry through consumption demand, capital allocation, and the technological innovation mechanism. The higher the consumption demand, the better the capital allocation and the technological innovation, and, meanwhile, the greater the role of innovation and development in promoting the intel-

ligent transformation of the local manufacturing industry. Third, the interaction coefficients of the indirect effects are −0.081, 0.187, and 0.576, respectively. The interaction coefficient between capital allocation, technological innovation, and Lnfin is positive. It is believed that the development of financial innovation can improve the efficiency of capital allocation and promote technological innovation, then promote the intelligent transformation and upgrading of the manufacturing industry in the surrounding areas [44]. Noninsignificant and negative consumption demand indicates that, in areas with higher consumption demand, financial innovation has a small promoting effect on the intelligent upgrading of the manufacturing industry in the surrounding areas and has a restraining effect. This may because financial innovation activities have certain externalities and spillovers which will stimulate the competition between regions. In addition to the scarcity of financial resources, the increase of financial resources in one region will lead to the decrease of ones in another region [2,27]. Based on prior literature [50], the profitability of financial capital will make capital flow to better-based areas. Although the consumption demand is high, it is restrained due to the lack of source power of manufacturing intelligence in the surrounding areas.

**Table 9.** Empirical results of mechanism test.

|  | Variable | Consumer Demand | Capital Allocation | Technological Innovation |
|---|---|---|---|---|
| Direct effect | lnfin | 0.196 *** (0.01) | 0.155 *** (0.00) | 0.672 *** (0.00) |
|  | lnmedium | 0.230 * (0.05) | 0.015 (0.62) | 0.260 *** (0.00) |
|  | lnmedium * lnfin | 0.225 ** (0.02) | 0.131 *** (0.00) | 0.063 ** (0.01) |
| Indirect effect | lnfin | 0.023 (0.86) | 0.021 (0.80) | 0.106 (0.15) |
|  | lnmedium | −0.033 (0.88) | 0.013 (0.86) | 0.105 (0.14) |
|  | lnmedium * lnfin | −0.081 (0.65) | 0.187 ** (0.04) | 0.576 * (0.08) |
| Total effect | lnfin | 0.219 * (0.08) | 0.176 ** (0.02) | 0.902 ** (0.01) |
|  | lnmedium | 0.197 (0.26) | 0.028 (0.69) | 0.119 (0.15) |
|  | lnmedium * lnfin | 0.144 (0.39) | 0.318 *** (0.00) | 0.147 ** (0.03) |

Note: ***, **, and * denote coefficients significant at 1%, 5%, and 10% statistical levels, respectively. The $p$ value is in brackets.

## 5. Conclusions and Implications

Based on the data of 30 provinces in China from 2008 to 2019, this paper examines the effect of financial innovation on the intelligent transformation and upgrading of the manufacturing industry. We first build a system for the intelligent transformation and upgrading of the manufacturing industry according to the existing literature research; then, we innovatively use the entropy weight method, DEA, and spatial econometrics to analyze the spatial correlation between financial innovation and the manufacturing industry, and further study its internal influence mechanism. The main conclusions are drawn as follows: (1) The level of intelligent upgrading of the manufacturing industry has been significantly improved, but the spatial distribution is uneven, showing the characteristics of a small gap between the east and a large gap between the central and western regions. This shows that the development of the manufacturing industry is uneven among regions. (2) According

to the spatial correlation test, financial innovation and the intelligent upgrading of the manufacturing industry has a strong spatial correlation, and there is a positive spatial spillover effect. This means that financial innovation can not only promote the intelligent transformation of the manufacturing industry in the region, but also promote the surrounding areas, and the spillover effect within the region is greater than that between regions. After the endogenous test, it further verifies the robustness of the conclusion that financial innovation contributes to the improvement of manufacturing intelligence in the spatial dimension. (3) The mechanism test indicates that the improvement of consumer demand, the optimization of capital allocation efficiency, and technological innovation are important transmission mechanisms for financial innovation to affect the intelligent upgrading of the manufacturing industry.

Financial innovation is an important carrier to promote the transformation and upgrading of the manufacturing industry. Through theoretical and empirical analysis, we clarify the impact and transmission path of financial innovation on manufacturing upgrading. The research conclusions not only help to strengthen and enrich the positive theoretical research on financial innovation supporting the intelligent development of manufacturing industry, but also provides a positive reference for promoting the coordinated development of the manufacturing industry among regions. Based on the previous results, several policy recommendations are made as a practical reference for policymakers. First, the financial industry should develop in harmony with the manufacturing industry, and we should attach great importance to the strong correlation between financial innovation and manufacturing intelligence. The manufacturing industry is the main body of the national economy, and its high-quality development is of great significance for building a new development pattern. Finance will play a more and more important role in many key links, such as manufacturing technology breakthrough, transformation and upgrading, industrial chain development, and global cooperation. Only by constantly accelerating innovation can financial institutions improve the adaptability, competitiveness, and inclusiveness of manufacturing finance, and strive to help the development of the manufacturing industry with high-quality financial services. The two are mutually reinforcing and inseparable. Therefore, the government departments must promote the coordinated development of the financial industry and the manufacturing industry and insist that the financial industry serve the real industry. Second, provincial innovation coordination and cooperation should be strengthened so to promote the development of intelligent manufacturing agglomeration. Due to the positive spatial correlation, the impact of manufacturing development between adjacent regions is obvious. Therefore, all provinces should make full use of the regional advantages brought by financial innovation and development to attract advanced manufacturing industries into the region. The implementation of a regional coordinated development strategy can better realize the organic unity of financial innovation and manufacturing intelligence, which is of great significance to narrow the regional gap and fundamentally solve the imbalance and disharmony. Third, at the regional level, consumption demand, capital allocation, and technological innovation play a part of the mechanism role in the impact of financial innovation on the intelligent transformation and upgrading of the manufacturing industry. Especially in the digital economy era, government departments should give full play to the role of these mechanisms to provide impetus for the development of the manufacturing industry. In these ways, the financial sector can avoid "disenchantment from reality to emptiness", thus guiding the high-quality development of the manufacturing industry.

Although this paper empirically tests the spatial effect of financial innovation on the intelligent transformation and upgrading of the manufacturing industry, it has positive theoretical and practical significance. However, this study still has some limitations. Limited by the availability, integrity, and continuity of data, we only use 15 three-level indicators to measure the intelligent transformation and upgrading level of the manufacturing industry. Future research can build a more reasonable, perfect, and index-rich measurement index system to obtain more accurate results. Moreover, other mechanisms between financial innovation and manufacturing intelligence can be expanded in the next step.

**Author Contributions:** Conceptualization, J.Z. and F.C.; methodology, F.C.; formal analysis, J.Z. and W.W.; investigation, J.Z.; data curation, F.C.; writing—original draft preparation, J.Z. and F.C.; writing—review and editing, J.Z., F.C. and W.W.; funding acquisition, J.Z. All authors have read and agreed to the published version of the manuscript.

**Funding:** This research was supported by the general program of the National Natural Science Foundation of China (grant number: 72171162), the major decision-making consulting program of Shanxi provincial government (grant number: zb20210805), Special project of science and technology strategy research in Shanxi Province (grant number: 202104031402090), the philosophy and social science planning project of Shanxi Province (grant number: 2021yy194), and the graduate education innovation project of Shanxi Province (grant number: 2021y636).

**Informed Consent Statement:** Not applicable.

**Data Availability Statement:** Not applicable.

**Conflicts of Interest:** The authors declare no conflict of interest.

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
