# Peer review of "The Spatial Effect of Financial Innovation on Intellectualized Transformational Upgrading of Manufacturing Industry: An Empirical Evidence from China"

_sustainability, doi:10.3390/su14137665_

Round 1

Reviewer 1 Report

Thank you very much for an opportunity to read your paper. Generally, I liked your paper, however, I’ve got some concerns:

1) I would suggest that in the Introduction authors not only report the previous research but emphasize what what done and what's lacking in the scientific literature which could support the relevance of the research.

2) To my opinion, all Hypothesis need deeper argumentation, i.e. Hypothesis 1 the part "Financial innovation can effectively promote..." it is not explained what authors mean by "effective promotion", or Hypothesis 3 it is mostly stated that " it produces spatial spillovers to the surrounding areas" but is not very clear what are channels or mechanizms which generate spatial impact of financial innovation.

3) Also it is not clear what authors consider to be a financial investment applied in manufacturing industry. It could be explained in more detailed. In the model description Financial innovation is proxied as respectively represent the financial capital stock of province but how we can consider it as innovation? The same is with the dependent variable - Intellectualized Transformation. 

4) As authors use Index for dependent variable how they avoid of doubling of the variables which are included as independent as well, i.e. Personnel input - as Human capital level and etc.

5) Research results could be supported with deeper interpretation and comparison with previous research. Suggestions are poorly related to the research. 

Author Response

We are grateful for the important comments of reviewers, we have modified our manuscript in details according to reviewers’ comments. Main modifications were emphasized through using red color fonts in the revised manuscript. For the Reviewers’ comments, we have given specific answers.

Reviewer 2 Report

A very well thought out and executed research. For each of the issues raised in the study a research hypothesis was proposed, which was then verified in the course of analysis using statistical methods.

However, some doubts may be raised:

1) the lack of a discussion section, in which it would be possible to compare the results of the conducted analyses with the studies made by other authors

2) too poor literature used in the paper, which mostly covers the works of Chinese authors, omitting the studies carried out in other regions, although in several places in the paper the Authors state: "There are many studies on this aspect in the existing literature"

3) the lack of clearly defined aim of the work, which seems to be necessary in case of multithreading of the conducted research

4) the results section focuses especially on the discussion of statistic dependencies, in principle limiting the analysis of spatial conditions to a minimum, e.g. it would be interesting problem, in what relation the research results are to Hu line

5) figures 2 and 3 in the present form are hardly legible

6) the language of the paper needs substantial corrections in terms of the style of narration (e.g. Some believe positive effect..., While, some believe no effect or negative)

Author Response

(The authors gave the same response as above.)

Reviewer 3 Report

The authors study spatial effect of financial innovation on intellectualized transformation upgrading of the manufacturing industry by using Chinese provincial panel data from 2008 to 2019. Overall, this paper is interesting. However, I suggest some comment as follows:

1. Please should revise the subsection 2.1. Financial innovation theory. Please explain Financial innovation theory carefully.

2. Please increase the sharpness of the figure 2 and figure 3.

3. Please position the pictures and tables properly.

4. Please focus advantage of this work.

Author Response

(The authors gave the same response as above.)

Round 2

Reviewer 1 Report

Dear Authors,

Thank you very much for your responses and made improvements.